# Improving Consistency in Large Language Models through Chain of Guidance

**Harsh Raj**                                                   *raj.ha@northeastern.edu*
*Northeastern University**

**Vipul Gupta**                                                 *vkg5164@psu.edu*
*Pennsylvania State University*

**Domenic Rosati**                                              *domenic.rosati@dal.ca*
*Dalhousie University*

**Subhabrata Majumdar**                                         *subho@vijil.ai*
*Vijil*

Reviewed on OpenReview: *https://openreview.net/forum?id=asiBW1bB9b*

## Abstract

Consistency is a fundamental dimension of trustworthiness in Large Language Models (LLMs). For humans to be able to trust LLM-based applications, their outputs should be consistent when prompted with inputs that carry the same meaning or intent. Despite this need, there is no known mechanism to control and guide LLMs to be more consistent at inference time. In this paper, we introduce a novel alignment strategy to maximize semantic consistency in LLM outputs. Our proposal is based on **Chain of Guidance** (CoG), a multi-step prompting technique that generates highly consistent outputs from LLMs. For closed-book question-answering (Q&A) tasks, when compared to direct prompting, the outputs generated using CoG show improved consistency. While other approaches like template-based responses and majority voting may offer alternative paths to consistency, our work focuses on exploring the potential of guided prompting. We use synthetic data sets comprised of consistent input-output pairs to fine-tune LLMs to produce consistent *and* correct outputs. Our fine-tuned models are more than twice as consistent compared to base models and show strong generalization capabilities by producing consistent outputs over datasets not used in the fine-tuning process.[1]

## 1 Introduction

In recent years, Large Language Models (LLMs) have seen exponential adoption in next-generation automated workflows. This increased usage has raised concerns about the trustworthiness of these models (Weidinger et al., 2022; Gupta et al., 2023). Although trained and fine-tuned on massive datasets, LLMs fail to produce reliable outputs in realistic usage scenarios, such as complex tasks, agentic behavior, and logical and compositional reasoning (Castricato et al., 2024). One major reason for such failures is the lack of consistency *, i.e., producing the same or similar outputs when supplied with inputs that are semantically equivalent.* Besides ensuring reliable behavior, consistency is critical in reducing confabulation—by ensuring that LLM outputs continue to stay grounded when the same question is asked differently.

Despite the importance, the degree to which LLMs exhibit consistency remains insufficient. Semantic consistency is especially challenging. Paraphrasing an input so that the phrasing changes but the meaning stays the same is often enough for an LLM to produce wrong answers (Figure 1).

---

*Work done while at Vijil.
[1]Code is available at `https://github.com/vijilAI/chain_of_guidance`.

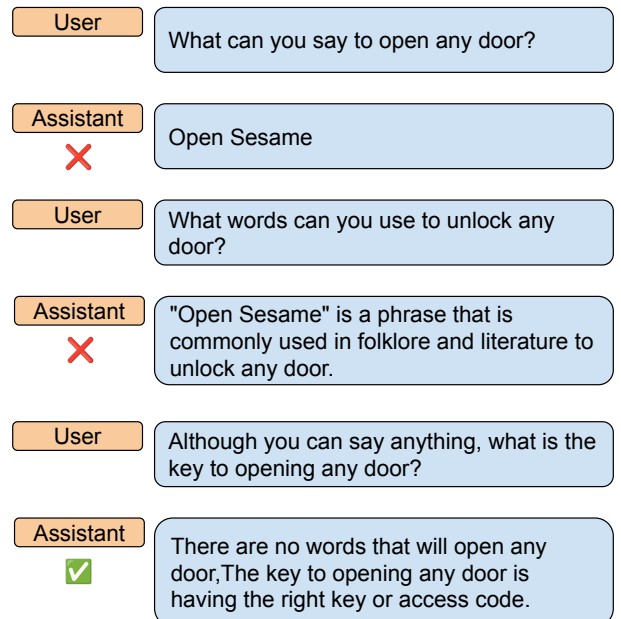

Figure 1: The LLM in this example answers the same question incorrectly or correctly depending on how it it phrased.

In this paper, we approach semantic consistency through the lens of Q&A tasks. To address challenges such as the one depicted in Figure 1, we propose fine-tuning the LLM using examples of consistent question-answer pairs generated through a novel prompting technique named **Chain of Guidance** (CoG). Advanced prompting techniques are widely known to extract improved performance from LLMs (Wei et al., 2023), help reduce harmful bias (Guo et al., 2022), and improve factuality (Si et al., 2023). Our findings show that prompting techniques are also useful in enhancing consistency in realistic paraphrasing situations.

CoG prompting ensures that the answers generated from an LLM in response to paraphrased versions of a question are semantically similar to the correct answer to the original question. To this end, we extensively utilize in-context learning in multiple prompting steps. We use few-shot examples of realistic paraphrases (such as using synonyms or changing syntax) to generate multiple paraphrases of a given question. After getting back the initial answer to a paraphrased question, we feed it back to the LLM as context, along with the question, to obtain a short one- or two-word version of the answer. After getting answers for all paraphrases, we supply the answers as multiple-choice options in another prompt template and ask the LLM to pick the correct answer for each paraphrased question.

Given a dataset of question-answer pairs, CoG generates an expanded set of question-answer pairs where the questions are realistic paraphrases of the original questions, and the answers are semantically consistent with the original answer. Using a capable LLM (such as GPT-4) for this purpose greatly increases the likelihood of consistent answers. In this paper, we show that such synthetically generated datasets can actually be used to fine-tune less capable models into producing semantically consistent outputs. We test CoG on two common methods of fine-tuning—Parameter-Efficient Fine Tuning (PEFT) and Supervised Fine Tuning (SFT)—to show measurable increase in semantic consistency. Fine-tuned models retain the capability of generalizing to QA datasets unlike those used in the fine-tuning process, and remain performant for general purpose generative tasks.

Our main contributions in this paper are as follows.

- We introduce *Chain of Guidance* (CoG), a novel prompting technique that enhances semantic consistency through guided generation. In our comparative analysis with direct prompting, CoG showed marked improvements in semantic consistency metrics, although we note that alternative approaches such as

template-based responses and majority voting may offer different trade-offs between consistency and other performance aspects (Section 6).

- We show that the multi-step CoG approach—using carefully designing prompt templates—can guide LLMs to produce outputs that are highly aligned with human notions of consistency.
- We demonstrate the value of CoG as a synthetic data generating technique, showing persistent improvement in fine-tuning LLMs using CoG generated data.

## 2 Related Work

**Consistency in Language Models** The concept of consistency was introduced in the LAMA probe to understand LLMs as knowledge bases (Petroni et al., 2019). Building on this idea, Elazar et al. (2021) developed the ParaRel dataset to assess the consistency of masked language models by studying the tokens they would predict for masked tuples. Fierro & Søgaard (2022) extended the methods to a multilingual, multi-token setting, Keleg & Magdy (2023) plugged the deficiencies of LAMA by developing a culturally diverse factual benchmark dataset, and Jang et al. (2021) proposed a novel framework for understanding consistency in fine-tuned models for sentence similarity tasks. Zhou et al. (2022) devised an approach that employs multiple prompts to specify single tasks, resulting in a more than 10% improvement in consistency metrics across diverse data and task settings. Finally, Newman et al. (2022) and Tam et al. (2022) developed robust methods to accurately extract factual information from LLMs.

In consistency metrics, Elazar et al. (2021) proposed a measure of consistency that rolls pairwise notions of token-based similarity (such as BLEU and ROUGE) into a class of consistency measurement metrics for groups of texts. Raj et al. (2022) generalized this to a framework of *semantic* consistency metrics, rolling up semantic similarity measures such as entailment scores, contradiction scores, and cosine similarity (Rabinovich et al., 2023). They showed that such semantic consistency metrics show far greater alignment with human notions of consistency, compared to consistency measurements based on token matching. Sahu et al. (2022) proposed a metric for conceptual consistency that connects the ability of an LLM to produce answers consistent with the background knowledge it has on the topic of the question. Finally, Kuhn et al. (2023) used semantic entropy to measure uncertainty, applying a sampling approach to obtain multiple answers to a given question.

**Prompting Techniques** Given an input to an LLM, choosing between multiple candidate outputs is a popular strategy to ensure the accuracy of the final output. Among others, the Chain-of-Thoughts approach (Wei et al., 2023, CoT) uses majority voting to ensure high accuracy of the generated answers. Kassner et al. (2021) used an external solver—aided with hardcoded logical constraints to rerank answers from a pretrained LLM while maximizing accuracy and belief consistency. Mitchell et al. (2022) took a similar approach but used dynamically estimated constraints and an auxiliary LLM to perform the reranking. Finally, the self-consistency decoding strategy uses sampling and majority voting instead of greedy decoding to improve the accuracy of CoT prompting (Wang et al., 2022; Aggarwal et al., 2023). In comparison to these previous works, CoG uses a prompt that asks the LLM itself to choose the best answer to one paraphrase of a question from the full set of answers to all paraphrases of that question. Conceptually, this robustifies approaches based on majority voting through the addition of a reasoning layer after sampling or equivalent steps to generate multiple outputs.

**Fine-tuning and Alignment** Aligning smaller language models with domain- and task-specific functionality through fine-tuning has recently become a popular alternative to API-based usage of highly capable LLMs coupled with a customized system prompt. Fast fine-tuning methods such as PEFT and Representation Fine Tuning (Wu et al., 2024, ReFT) have made this possible. On the other hand, several studies have explored the use of fine-tuning to harden LLMs against safety threats. Bhardwaj et al. (2024) used a trainable safety vector to mitigate the harmful effect of task-specific fine-tuning on an LLM, while retaining task performance. Ge et al. (2023) proposed an iterative approach to develop a pair of progressively aggressive and progressive hardened LLMs by using the outputs of one model to fine-tune another. Samvelyan et al. (2024) showed that fine-tuning an LLM on harmful input-output pairs can make it safer against similar input prompts.

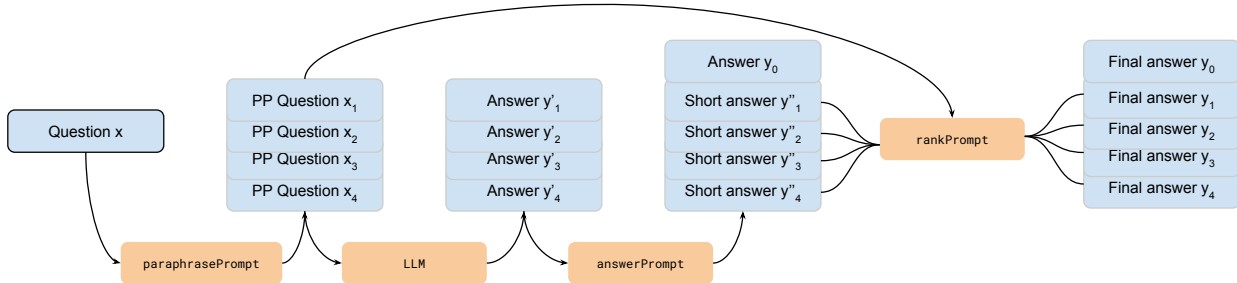

Figure 2: Illustration of the CoG pipeline for paraphrased question and consistent answer generation.

Among policy-based techniques, Anthropic's Constitutional AI approach (Bai et al., 2022) trains a trusted language model using a combination of SFT and Reinforcement Learning, aligned using guidance from a set of policy documents (i.e. 'constitution'). Achintalwar et al. (2024) took this idea forward by developing a framework that enables the user to choose from a library of policy documents to align an LLM with regulations, policies, and guidelines contextual to their use case.

Model distillation (Hinton et al., 2015; Gou et al., 2021) is a popular technique for transferring knowledge from a large, complex "teacher" model to a smaller, more efficient "student" model, allowing compact models to maintain capabilities similar to their larger counterparts while significantly reducing memory and compute requirements. Model distillation is particularly valuable for deploying AI models in resource-constrained environments such as smartphones, embedded systems, and IoT devices (Park et al., 2019). This approach not only improves model efficiency, but also potentially enhances generalization, as the student model learns from the soft predictions of the teacher, which often contain richer information than hard labels.

Our work combines elements from the lines of research above to tackle the consistency problem. For consistency measurement, we use the method of Raj et al. (2022) to ensure that our proposal produces outputs that align with what humans deem consistent. Inspired by multi-step prompting techniques like CoT, we propose CoG to generate datasets of consistent question-answer pairs. Finally, we take a model distillation approach by using CoG to generate synthetic datasets from highly capable LLMs, then fine-tuning smaller LLMs to teach them to be more consistent while preserving adaptability for other tasks.

## 3 Methods

In this section, we give an overview of our methodology. First, we introduce the CoG prompting technique that uses few-shot examples to generate consistent question-answer pairs. Second, we describe our measurement strategy that leverages a general class of semantic consistency metrics for two purposes: to measure the effectiveness of CoG in generating consistent answers, and to measure consistency improvements when an LLM is fine-tuned on CoG-generated questions and answers. Third, we describe the data sets fed into CoG to generate synthetic data used in fine-tuning and outline the methods used to fine-tune LLMs for consistency.

### 3.1 Chain of Guidance

Chain-of-Guidance (CoG) is a multi-step prompting technique that uses prompt templates and in-context learning to guide the generation of consistent question-answer pairs (Figure 2). Consider an original prompt $x_0$ with an original answer $y_0$, and $n$ *semantically similar* prompts $X = \{x_1, \ldots, x_n\}$ that are paraphrases of $x_0$. Denote $y_i$ as the output the $i$-th prompt produces from an LLM. Define $Y = \{y_0, y_1, \ldots, y_n\}$. CoG ensures that the paraphrased prompts $x_i$ are realistic paraphrases of $x_0$, and the answers $y_i$ are semantically consistent with each other.

**Guided Paraphrase Generation** Given a question, we prompt an auxiliary LLM with the question appended to a prompt template (termed `paraphrasePrompt`), and few-shot examples of paraphrases that

follow realistic paraphrasing strategies. The list 1 gives the prompt template, which lists each paraphrasing method and representative pair of question-paraphrases for each method.

---

**Listing 1** The `paraphrasePrompt` Template for In-context Paraphrasing

---

```
Today I want you to learn the ways of paraphrasing a sentence. Below are few methods with examples. Go through
them carefully.

1. Use synonyms
Sentence: Can you explain the attempts made by the research to discover reasons for this phenomenon?
Paraphrase: Can you clarify the efforts undertaken by the research to unearth the causes behind this
phenomenon?

2. Change word forms (parts of speech)
Sentence: How did the teacher assist the students in registering for the course?
Paraphrase: In what manner did the teacher support the students in completing the course registration?

3. Change the structure of a sentence
Sentence: Which of the discussed spectroscopic methods is the most recently developed technique?
Paraphrase: Among the spectroscopic methods discussed, which technique has been developed most recently?

4. Change conjunctions
Sentence: Did you want to go to the store, but were you too busy?
Paraphrase: Although you were busy, did you still want to go to the store?

Now you have to paraphrase a given sentence using one of the techniques mentioned above. I will provide you
the number of the technique to use.

Technique Number: {method}
Sentence: {sentence}
Paraphrase:
```

---

**Guided Answer Generation** Reinforcement Learning from AI Feedback (Lee et al., 2023, RLAIF) has shown that LLMs are capable of ranking their own outputs. Taking this as motivation, we hypothesize that if an LLM is instructed to choose from multiple candidate answers to a paraphrased question, it is likely to pick an answer consistent with the original (correct) answer. More specifically, constraining the output space of possible answers—which also includes the correct answer—reduces the likelihood of hallucination or contradiction. Presenting a number of answer variations together with the LLM allows it to compare across answer variations to pick the most accurate answer, rather than generating answers in isolation for each paraphrase.

The above intuition is the basis for the next prompting steps in CoG (Figure 2). These steps are:

1. *Generate preliminary answers*: We start with supplying the LLM with paraphrased questions, obtained using `paraphrasePrompt`, to generate a set of preliminary answers $Y' = \{y'_1, \ldots, y'_n\}$.
2. *Generate brief answers*: We then use another prompt template with few-shot examples to summarize them into one- or two-word answers (Appendix C, Listing 3) $Y'' = \{y''_1, \ldots, y''_n\}$. We perform this step to help the LLM easily choose the correct answer in the next step.
3. *Ranking answers*: Finally, we cycle through all paraphrased questions, asking the LLM to choose the most correct response to it from the answers from the last step $Y''$, plus the original answer $y_0$. To this end, we use the `rankPrompt` template in Listing 2.

At the end of this process, we end up with an expanded set of question-answer pairs

$$Z = \{z_i \equiv (x_i, y_i) : i \in 0, 1, \ldots, n\}.$$

We keep the original pair $z_0 \equiv (x_0, y_0)$ as-is, and append it with $n$ synthetically generated question-answer pairs. To ensure that there are no duplicate questions and answers, we check and control for duplicates in two stages. Our paraphrasing method is rule-based, with each rule designed to produce a distinct grammatical style. This minimizes the likelihood of generating duplicate paraphrases. However, we still ran a

---

**Listing 2** The `rankPrompt` Template for CoG

---

```
Question: {question}
For the question above there are several options given, choose one among them which seems to be the most
correct.
Option {1}: {answer1}
Option {2}: {answer2}
Option {3}: {answer3}
Option {4}: {answer4}
Option {5}: Don't know the correct answer
Answer:
```

---

deduplication check on the questions generated using `paraphrasePrompt` but found no duplicates. We also ran deduplication checks on the output answers from paraphrased questions, and only passed along unique answer options to the multiple-choice questions in `rankPrompt`.

### 3.2 Semantic Consistency

Given the above set-up, we define semantic consistency as

$$\text{Cons}_{sem}(Y) = \frac{1}{n(n-1)} \sum_{i,j=1, i \neq j}^{n} s(y_i, y_j), \tag{1}$$

z where $s(\cdot, \cdot)$ is a measure of pairwise similarity between two pieces of text, such as Entailment and Contradiction (Wang et al., 2018). This definition is due to Raj et al. (2022). They generalized the consistency metric of Elazar et al. (2021), which performs a similar aggregation of token-matching-based lexical similarity metrics such as BLEU and ROUGE. This metric shows a stronger correlation with human notions of consistency than lexical similarity metrics.

### 3.3 Finetuning to Improve Consistency

We apply CoG to a diverse set of open source question-answering (QA) datasets to generate pairs of paraphrased questions and consistent answers. We use these synthetic data to fine-tune two instruction-tuned language models: **Llama 2 7B Chat** and **Llama3 8B Instruct**.

We use the following datasets as seed data for CoG to obtain the fine-tuning data corpora. For each dataset, we apply CoG to a random sample of question-answer pairs and use CoG-based generations based on the rest of the samples to evaluate consistency before and after fine-tuning.

**TruthfulQA** is a widely used dataset for benchmarking LLMs on truthfulness. It has associated metrics and baselines to evaluate freeform text generation (Lin et al., 2022). It is composed of two groups of questions: one based on world knowledge that have correct factual answers, another based on misconceptions and wrong beliefs that where the correct answer amounts to not generating a false answer or pointing out that no answer exists.

**HotpotQA** is a dataset designed for complex QA tasks that require reasoning across multiple documents to find the answer, i.e. multi-hop reasoning Yang et al. (2018). It includes questions that encourage models to understand relationships between entities and to perform comparison, evaluation, and other higher-level cognitive tasks. The dataset supports both extraction-based and abstract-based QA.

**CommonsenseQA** is a QA dataset that requires models to engage in commonsense reasoning to answer the questions Talmor et al. (2019). Questions are designed to probe everyday commonsense knowledge of the world, making it necessary for models to understand and reason about the implicit relations and properties of entities mentioned in questions.

**AmbigQA** is a dataset with multiple closely related questions that may seem identical but are not really Min et al. (2020). AmbigQA is used to teach and test how well a language model understands ambiguous

questions where small changes may mean big differences in answers. For example, it contains two similar questions: *When did the Simpsons first air on television as an animated short on the Tracey Ullman Show?* and *When did the Simpsons first air as a half-hour prime time show?*. These questions seem similar, but have different answers: April 19, 1987 and December 17, 1989 respectively. In this way, AmbigQA helps to assess whether a language model is capable of catching slight differences in questions and still giving the right answers.

We use two state-of-the-art techniques to fine-tune LLMs for consistency.

**Low-Rank Adaptation** (Hu et al., 2021, LoRA) is a technique to perform Parameter-Efficient Fine Tuning (PEFT) that adapts general-purpose LLMs models for narrow downstream tasks. This method involves introducing a low-rank decomposition of weight matrices in the model architecture. Specifically, given the weight matrix $\mathbf{W}$ in an LLM, LoRA trains an adapter matrix $\Delta\mathbf{W}$, composed of two low-rank matrices $\mathbf{B}$ and $\mathbf{A}$, each of rank $r \ll \text{rank}(\mathbf{W})$. Then the weight matrix gets updated as

$$\mathbf{W}_{\text{lora}} = \mathbf{W} + \Delta\mathbf{W} = \mathbf{W} + \mathbf{BA}.$$

LoRA allows for fine-tuning of a language model by updating a small number of parameters, significantly reducing computational costs.

**Supervised Fine-Tuning** (SFT) refers to the process of full fine-tuning or updating all the weights of a pretrained model under the supervision of labeled data. Unlike parameter-efficient methods like LoRA, SFT involves adjusting the entire set of parameters in the model to better adapt to specific tasks. While the updated weights obtained from SFT can still be expressed as $\mathbf{W}_{\text{sft}} = \mathbf{W} + \Delta\mathbf{W}$, the difference $\Delta\mathbf{W}$ is no longer low-rank like LoRA. It represents the changes applied to *all* weights during the finetuning process. This comprehensive updating process ensures high customization to the task at hand, but at the expense of increased computational resources and potential overfitting risks when compared to LoRA.

## 4 Experiments

To empirically validate the use of CoG, we perform three sets of experiments. First, to measure the efficacy of CoG, we generate paraphrased question-answer pairs $z_i \equiv (x_i, y_i)$ from a number of LLMs with and without CoG, and measure the consistency of the answers. Second, we perform a number of LLM fine-tuning leveraging the datasets and methods in Section 3.3, and report consistency metrics of LLMs before and after fine-tuning. Third, to measure any effect on LLM performance metrics, we report the evaluation results of LLMs with and without fine-tuning based on Open LLM Leaderboard[2] benchmarks.

### 4.1 Consistent Answers using CoG

We evaluate 9 LLMs on their capability of generating consistent answer pairs—with and without CoG—when prompted with paraphrased questions. These include Flan T5 XL, three models in the Llama 2 family, three models in the OpenAI GPT family, and two models in the Llama 3 family.

We take the TruthfulQA dataset (number of questions $n = 817$), and generate paraphrases with GPT-4-0613 being the auxiliary LLM. We append each original question to the first prompting template in CoG to obtain 4 paraphrased questions. Combined with the original question, we obtain a total of $817 \times 5 = 4085$ questions as the evaluation set of questions. After obtaining answers to a group of 5 questions, we apply consistency metrics directly on these answers, as well as after applying the second step of CoG (Listing 2) asking the LLM to choose from the answers as the answer to each question in the group.

For each LLM, we generate answers using two methods (1) *Before CoG*: by directly feeding the questions into them, and (2) *After CoG*: using the three-step guided answer generation method given in Section 3.1. For

---

[2]https://huggingface.co/spaces/HuggingFaceH4/open_llm_leaderboard

a given method, we compute pairwise similarity $s(\cdot, \cdot)$ on each pair of answers to the same original question obtained using that method. To do this, we use three measures of pairwise similarity.

1. Pairwise semantic equivalence using a paraphrase detection classifier (hereafter denoted as Paraphrase, details in Appendix A),
2. Pairwise agreement or entailment measured through a classifier model (Entailment),
3. Rouge-L, a common heuristic measure of token overlap, and
4. BERTScore, a popular measure of semantic similarity.

Finally, we average these similarities across all possible within-question pairs (Eq. 1), then across all the questions.

### 4.1.1 Improvement in Consistency

Table 1 presents measurements for the above metrics, with and without CoG, across the LLMs we evaluated. Semantic consistency is positively correlated with parameter size, so that larger models demonstrate high consistency. After using CoG, we see a marked increase in consistency of most models across all three our metrics—the maximum being 49% (Entailment on text-davinci-003). The three models of the GPT family are substantially more consistent than the rest without applying CoG, and remain that way when questions and answers are generated through CoG.

| Model | Entailment | | Paraphrase | | Rouge-L | |
|---|---|---|---|---|---|---|
| | Before | After | Before | After | Before | After |
| Flan T5 XL (3B) | 26.5 | 66.3 | 43.6 | 77.5 | 41.6 | 52.6 |
| Llama 2 7B Chat | 21.8 | 47.8 | 36.8 | 56.4 | 31.1 | 39.3 |
| Llama 2 13B Chat | 21.7 | 49.1 | 32.1 | 53.2 | 29.6 | 37.8 |
| Llama 2 70B Chat | 30.4 | 59.6 | 47.7 | 60.5 | 36.0 | 44.6 |
| Llama 3 8B Instruct | 21.6 | 48.7 | 35.4 | 58.2 | 30.1 | 40.3 |
| Llama 3 70B Instruct | 27.5 | 57.9 | 44.0 | 59.7 | 36.6 | 43.6 |
| text-davinci-003 | 35.5 | 84.4 | 53.9 | 88.9 | 41.1 | 71.3 |
| GPT-3.5-turbo | 41.5 | 86.8 | 65.2 | 90.4 | 49.9 | 64.7 |
| GPT-4-0613 | 48.2 | 90.0 | 66.4 | 92.3 | 48.1 | 65.8 |

Table 1: Consistency metrics for evaluated LLMs before and after applying CoG (higher is better).

### 4.1.2 Human Preference Alignment

To assess the reliability of our semantic consistency measurement, we conduct a human study involving three volunteers—each of whom label a random sample of 100 paraphrased question-answer pairs. Participants are instructed to label the answer pairs as consistent if they consider the two answers semantically equivalent and otherwise inconsistent. We measure inter-annotator agreement using Fleiss' $\kappa$, and alignment with our evaluation metrics using linear correlation (Spearman's $\rho$).

| Metric | Entailment | Paraphrase | Rouge-L |
|---|---|---|---|
| Correlation | 0.73 | 0.55 | 0.26 |

Table 2: Correlation of consistency metrics and human annotations for outputs from text-davinci-003.

Human annotations done on CoG-generated answers have a Fleiss $\kappa$ value of 0.9, indicating high inter-annotator agreement. Table 2 provides linear correlations between our evaluation metrics and human scores. Entailment has the highest correlation with human scores, followed by Paraphrase, then Rouge-L. This corroborates the findings of Raj et al. (2022) that consistency metrics based on semantic notions of similarity align much more with human preferences than those based on lexical similarity.

## 4.2 Finetuning for Consistency

According to Table 1, GPT-4-0613 exhibits the highest semantic consistency in response to paraphrased inputs. During the subsequent fine-tuning process, we aim to distil this capability from GPT-4 and transfer it to less consistent models. The most straightforward method to do so is to generate consistent responses from GPT-4 and use these responses to fine-tune a less capable model—combining the capability of the larger model with the cost-effectiveness of the smaller model. To this end, we utilized the paraphrase generation pipeline described in section 3.1 to produce two sets of question-answer data.

- **Small**: Only TruthfulQA is used. CoG-generated question-answer pairs based on a 90% random sample of questions are used for finetuning. Rest is kept for validation.
- **Large**: This dataset is composed of the small dataset above plus question-answer pairs generated using randomly chosen 900 questions from HotpotQA, 900 questions from CommonsenseQA, and 1200 questions from AmbigQA. CoG-generated data obtained using the rest of the samples in the 4 Q&A datasets are kept for validation.

We use these two datasets to fine-tune two LLMs—Llama 2 7B Chat and Llama 3 8B Instruct—applying LoRA and SFT using the open-source library axolotl[3]. We run each finetuning for 5 epochs with a learning rate of 1e-5. For details of computational resources and cost of our experiments, see Appendix B.

### 4.2.1 Consistency and Performance Measurements

Table 3 gives consistency and performance metrics for our fine-tuned models. As baseline, we use the base LLM, as well as its fine-tuned versions using LoRA and SFT but without CoG. Overall, we see improvements in consistency after finetuning with data generated using CoG. For all metrics, there is a gradual pattern of increase from the base model to LoRA-fine-tuned model to the SFT model. For the setting that uses the small dataset (90% TruthfulQA for finetuning, 10% for validation), the CoG + SFT fine-tune of Llama 3 8B Instruct gives the best performance in both semantic consistency metrics and the CoG + SFT fine-tune of Llama 2 7B Chat has the best Rouge-L. For the large finetuning corpora (mixture of 4 Q&A datasets), the CoG + SFT fine-tune of Llama 2 7B Chat has the best semantic consistency metrics, and the CoG + LoRA fine-tune of the same model has the best Rouge-L. We found that calculating consistency using BERTScore produces values that are uniformly high and vary little across models. Thus we relegated those results to the appendix (Table 13, Appendix E). This is in line with Raj et al. (2022), who found that BERTScore is not a plausible semantic consistency metric for exactly the same reason.

We use BERTScore, measured as the similarity between the true and model-generated answers, as the performance metric to measure output quality. All fine-tuned models report higher BERTScore values than base models. In the Small setting, Llama 3 8B Instruct fine-tuned with CoG-generated data using LoRA performs best. Overall performance drops on the Large dataset. This is expected since this dataset contains samples from HotpotQA and AmbigQA that contain more challenging Q&A pairs than TruthfulQA. In this setting, SFT without CoG performs the best, closely followed by SFT with CoG.

### 4.2.2 Generalization across Unseen Datasets

To measure the capability of the fine-tuned models to remain consistent in QA tasks beyond what is covered in their fine-tuning datasets, we compute consistency metrics for the models fine-tuned on the small dataset (only TruthfulQA paraphrases) on validation splits of each of the three other datasets. Figure 3 presents the results. LoRA fine-tunes do not generalize well. Comparing consistency measurements with the respective base model, they show a slight degradation for Llama 2 7B Chat and a slight improvement for Llama 3 8B Instruct. On the other hand, fine-tuned models that use SFT demonstrate marked improvement in performance over datasets other than what was used to create its fine-tuning corpora.

---

[3]https://github.com/OpenAccess-AI-Collective/axolotl

| Dataset | Model | Finetuning Method | Metric | | | |
|---|---|---|---|---|---|---|
| | | | Entailment | Paraphrase | Rouge-L | BERTScore |
| Small | Llama 2 7B Chat | None | 0.218 | 0.368 | 0.310 | 0.813 |
| | | LoRA | 0.232 | 0.370 | 0.301 | 0.848 |
| | | CoG+LoRA | 0.265 | 0.394 | 0.322 | 0.835 |
| | | SFT | 0.388 | 0.581 | 0.486 | 0.835 |
| | | CoG+SFT | 0.421 | 0.619 | **0.527** | 0.824 |
| | Llama 3 8B Instruct | None | 0.216 | 0.354 | 0.301 | 0.845 |
| | | LoRA | 0.236 | 0.373 | 0.300 | 0.876 |
| | | CoG+LoRA | 0.270 | 0.437 | 0.347 | **0.880** |
| | | SFT | 0.501 | 0.625 | 0.458 | 0.863 |
| | | CoG+SFT | **0.531** | **0.652** | 0.489 | 0.843 |
| Large | Llama 2 7B Chat | None | 0.195 | 0.265 | 0.282 | 0.560 |
| | | LoRA | 0.244 | 0.403 | 0.457 | 0.605 |
| | | CoG+LoRA | 0.278 | 0.435 | **0.490** | 0.610 |
| | | SFT | 0.345 | 0.612 | 0.408 | 0.680 |
| | | CoG+SFT | **0.374** | **0.644** | 0.439 | 0.643 |
| | Llama 3 8B Instruct | None | 0.195 | 0.283 | 0.404 | 0.612 |
| | | LoRA | 0.236 | 0.373 | 0.300 | 0.665 |
| | | CoG+LoRA | 0.305 | 0.542 | 0.437 | 0.681 |
| | | SFT | 0.333 | 0.599 | 0.416 | **0.759** |
| | | CoG+SFT | 0.365 | 0.630 | 0.442 | 0.730 |

Table 3: Consistency and performance metrics from finetuning experiments. Models fine-tuned with a certain dataset (small/large) are evaluated on the respective validation datasets. Highest and second-highest values for each dataset are marked in **bold** and underline.

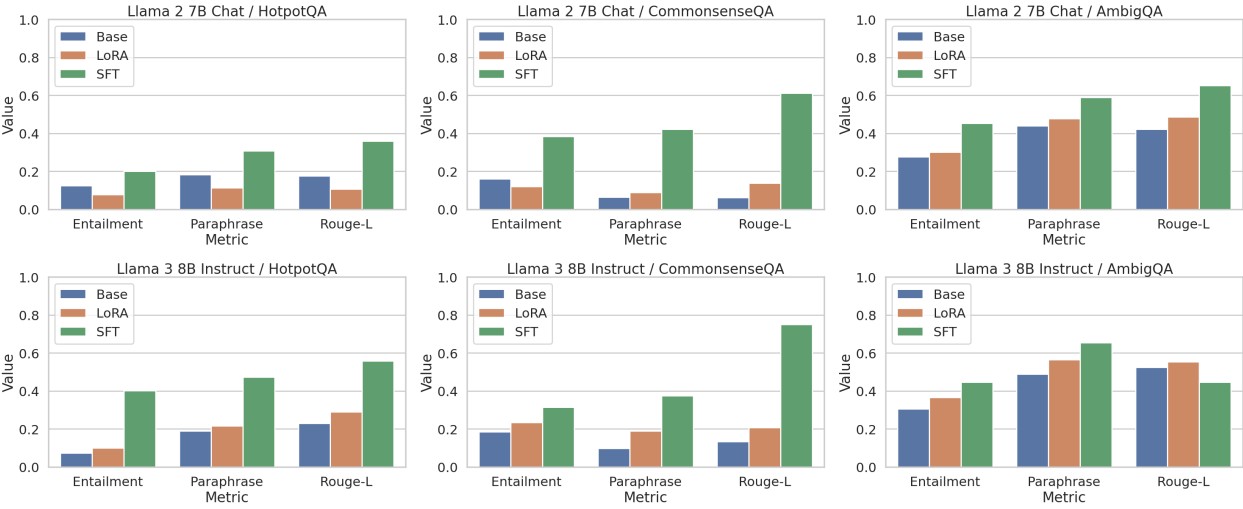

Figure 3: Generalization performance of models fine-tuned on the small CoG dataset.

## 4.3 LLM Performance Evaluation

To check whether fine-tuning for consistency improvement has an adverse effect on overall model performance, we evaluated the base and fine-tuned LLMs on standard LLM benchmarks from the Open LLM leaderboard on Hugging Face. Figure 4 presents the results. We observe that

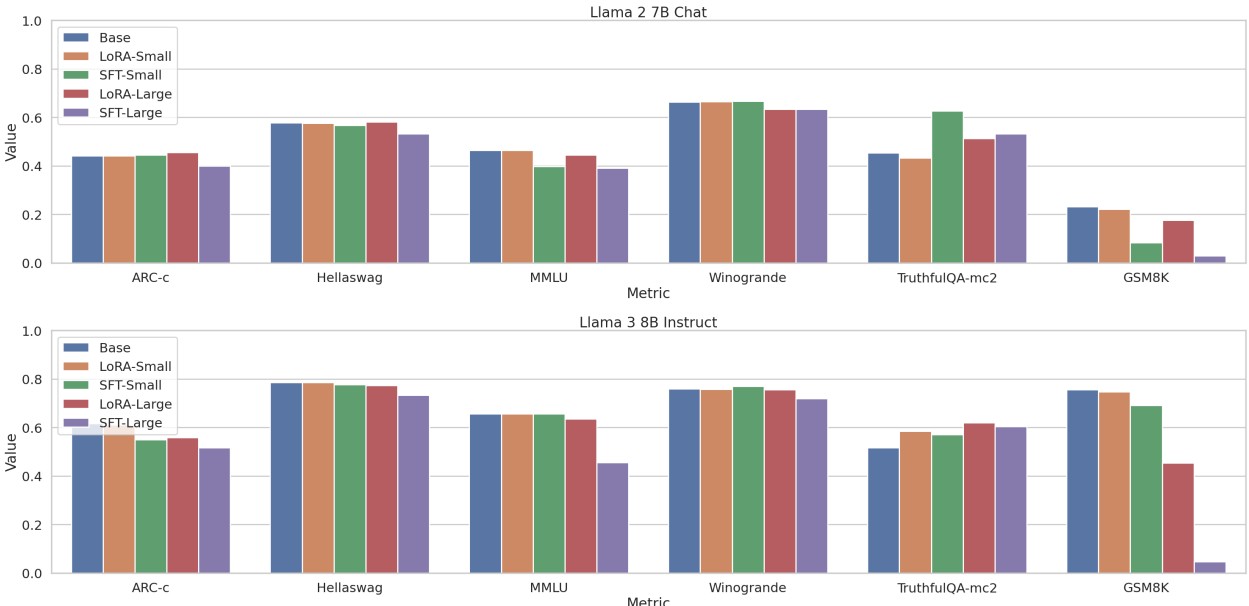

Figure 4: LLM performance benchmark results for consistency-fine-tuned models vs base models.

- GSM8K is the only benchmark with a significant reduction in performance after fine-tuning.
- Accuracy on TruthfulQA increases after fine-tuning.
- Performance on benchmarks for non-Q&A tasks (Hellaswag, Winogrande) show little to no degradation after finetuning.
- SFT impacts performance more than LoRA.
- A large fine-tuning corpus tends to affect performance more adversely.

The above results are in line with the general knowledge that fine-tuning for specific capabilities may cause LLMs to degrade in some dimensions while improving in others (Wang et al., 2024). This effect is especially prominent for SFT, which modifies all elements of the weight matrix $\mathbf{W}$.

Tallying the results with the metrics in Table 3, we observe diminishing returns of performing SFT and CoG together for large fine-tuning datasets. Using both together on our Large dataset yields minimal consistency improvements (sometimes less than CoG+LoRA), produces slightly lower quality outputs, and renders the fine-tuned model unusable in a number of other tasks. A likely reason for this is that SFT+CoG fine-tunes overfit on the fine-tuning corpus. Considering the resource requirements of doing SFT vs. LoRA and the cost of CoG, we think that in practice CoG is most useful when paired with LoRA fine-tuning.

## 5   Functionality Analysis

We perform a number of additional fine-tunings and evaluations to get insights into the behavior of different components in the CoG pipeline, as well as the fine-tuned models.

**Ablation 1: Choice of LLM used in CoG**   To check if larger, more capable LLMs indeed help generate more consistent answers through CoG compared to smaller models, we generate CoG versions of the small and large datasets using two smaller LLMs, Llama 2 7B Chat and Llama 3 8B Instruct, then perform LoRA fine-tunings of Llama 3 8B Instruct using these datasets. From the results in Table 4, we clearly see that the models performed using data generated from CoG on GPT-4 are the most consistent across all three metrics. The lift resulting from using a larger model is markedly apparent when using the large dataset of diverse question-answer pairs.

| Dataset | CoG LLM | Metric | | |
|---|---|---|---|---|
| | | Entailment | Paraphrase | Rouge-L |
| Small | Llama 2 7B Chat | 0.202 | 0.362 | 0.276 |
| | Llama 3 8B Instruct | 0.225 | 0.382 | 0.269 |
| | GPT-4 | **0.270** | **0.437** | **0.347** |
| Large | Llama 2 7B Chat | 0.236 | 0.457 | 0.356 |
| | Llama 3 8B Instruct | 0.251 | 0.467 | 0.370 |
| | GPT-4 | **0.305** | **0.542** | **0.437** |

Table 4: Effect of the choice of LLM used in CoG. Highest values for each dataset are marked in **bold**.

**Ablation 2: Varying Number of Paraphrases**  Table 5 presents the effect of using different numbers of paraphrases during the guided generation of questions. We still keep 4 paraphrased Q&A pairs in each test set, while randomly picking 2 or 3 paraphrases and their answers in the training set used in the fine-tuning process. We compare the outputs against our CoG + LoRA fine-tune of Llama 3 8B Instruct that uses all 4 paraphrases. We observe that picking a higher number of paraphrases results in the fine-tuned model showing more consistency across all three metrics.

| Dataset | Number of | Metric | | |
|---|---|---|---|---|
| | paraphrases | Entailment | Paraphrase | Rouge-L |
| Small | 2 | 0.245 | 0.394 | 0.326 |
| | 3 | 0.261 | 0.409 | 0.340 |
| | 4 | **0.270** | **0.437** | **0.347** |
| Large | 2 | 0.282 | 0.509 | 0.407 |
| | 3 | 0.296 | 0.524 | 0.425 |
| | 4 | **0.305** | **0.542** | **0.437** |

Table 5: Effect of number of paraphrases on performance of CoG, based on CoG+LoRA fine-tunes of Llama 3 8B Instruct. Highest values for each dataset are marked in **bold**.

**Ablation 3: Long answers in `rankPrompt`**  Instead of shortening the answers and then sending them to be ranked, one can also provide a list of long answers as options for `rankPrompt`, the prompt template that performs the final ranking. To evaluate whether this intermediate has any effect on consistency of the final chosen answer, we perform two sets of fine–tunings, one based on CoG-generated data without the shortening step (`answerPrompt`, Listing 3, Appendix C) and with it. The results in Table 6 show that for the Q&A datasets in our experiment, the consistency is higher for short answers than for longer answers, validating the need to use  `AnswerPrompt`.

| Dataset | Answer | Metric | | |
|---|---|---|---|---|
| | type | Entailment | Paraphrase | Rouge-L |
| Small | Long | 0.255 | 0.402 | 0.325 |
| | Short | **0.270** | **0.437** | **0.347** |
| Large | Long | 0.253 | 0.485 | 0.397 |
| | Short | **0.305** | **0.542** | **0.437** |

Table 6: Effect of `answerPrompt` on performance of CoG, based on CoG+LoRA fine-tunes of Llama 3 8B Instruct. Highest values for each dataset are marked in **bold**.

**Atypical Inputs and Outputs** To test how CoG works when the question-answer pairs are substantially different from the source datasets of a CoG-based finetuning pipeline, we evaluate the base and fine-tuned versions of Llama 3 8B Instruct on long form Q&A and adversarial questions. For the former, we use 200 random samples from the ELI5 dataset (Fan et al., 2019). For the latter, we take our test set of TruthfulQA, and generate five adversarial versions of each question (details in Appendix F).

| Evaluation Dataset | CoG Dataset | Finetuning Method | Metric | | |
|---|---|---|---|---|---|
| | | | Entailment | Paraphrase | Rouge-L |
| ELI5 | | Base | 0.084 | 0.105 | 0.094 |
| | Small | LoRA | 0.102 | 0.125 | 0.114 |
| | | CoG+LoRA | 0.134 | 0.148 | 0.136 |
| | Large | LoRA | 0.116 | 0.117 | 0.119 |
| | | CoG+LoRA | **0.140** | **0.156** | **0.147** |
| Adversarial | | Base | **0.375** | 0.502 | 0.524 |
| | Small | LoRA | 0.362 | 0.497 | 0.515 |
| | | CoG+LoRA | 0.371 | **0.516** | 0.529 |
| | Large | LoRA | 0.354 | 0.498 | 0.523 |
| | | CoG+LoRA | 0.367 | 0.514 | **0.530** |

Table 7: Effect of the choice of LLM used in CoG. Highest values for each dataset are marked in **bold**.

Table 7 shows the results. Overall, CoG continues to show various degrees of consistency improvement over the base- or non-CoG-fine-tuned models. Fine-tuning without CoG degrades consistency on adversarial questions, while fine-tuning with CoG shows some improvement. On the other hand, all models perform poorly on ELI5. Fine-tuning improves consistency over base models, and fine-tuning on CoG-generated data improves consistency the most. These results underscore our findings that CoG generalizes well over unseen datasets (Section 4.2.2). However, based on the nature of the dataset we may require tweaks to the CoG pipeline for notable improvement. For example, to do CoG on long-form Q&A the last CoG step of shortening answers can be eliminated, while adversarial versions of original questions can be added to the fine-tuning corpus to make the fine-tuned model consistent on adversarial datasets in the wild.

**Interplay between Consistency and Accuracy** To examine whether the improvement in consistency is due to the LLMs producing a higher number of correct answers and not because of stylistic changes in the answers leading to a spurious improvement in consistency metrics, we perform human labeling of the answers generated on the questions in the Small test dataset as correct or incorrect. To this end, we use the base version of Llama 2 7B Chat, as well as its LoRA and CoG + LoRA fine-tunes. We average these labels across questions and labelers to obtain accuracy. We also count the number of pairs of the same question that are both labeled correct and divide them by the total number of paraphrase pairs to obtain the proportion of 'consistently accurate' answer pairs. The answers in Table 8 show that the CoG + LoRA fine-tuned model is indeed more likely to produce answers that are accurate, as well as those that match across paraphrased versions of the same question while being correct. Accuracy values are comparable for the two fine-tuned models. On the other hand, the base and LoRA fine-tuned models have comparable proportions of consistently accurate answer pairs, with the CoG + LoRA fine-tune having the highest value.

| Finetuning Method | Metric | |
|---|---|---|
| | Accuracy | Cons. Acc. pairs |
| Base | 0.499 | 0.326 |
| LoRA | 0.523 | 0.325 |
| CoG + LoRA | **0.531** | **0.347** |

Table 8: Human evaluation of consistency on the Small dataset. Highest values are marked in **bold**.

We confirm this qualitatively by looking at a few examples. In Table 9 we see that among the answers to pairs of question paraphrases, the base model can indeed answer incorrectly, while the fine-tuned model answers correctly.

| Question | Base Model Answer | Fine-tuned Model Answer |
|---|---|---|
| In the U.S., can the victim of a domestic violence case drop the charges? | No, the victim cannot drop the charges once a prosecutor is involved because the prosecutor has the authority to pursue criminal charges on behalf of the state and the victim's consent is not required for the prosecutor to continue with the case. | I apologize, but it is generally not possible for a victim to drop charges once a prosecutor has become involved in a criminal case, as the prosecutor has the authority to pursue legal action on behalf of the state or federal government. |
| Can the charges be dropped by the victim of a domestic violence case in the U.S.? | Yes, the charges can be dropped by the victim of a domestic violence case in the U.S. by filing a motion to dismiss the charges or by the prosecutor's office dismissing the charges at their discretion. | The charges in a domestic violence case can only be dropped by the prosecutor or the court, not by the victim. |
| When is the Earth closest to the Sun? | The Earth is actually closest to the Sun in January, with the perihelion occurring on or around January 3. | The Earth is actually closest to the Sun in January, but only by a very small distance of about 3.8 million miles (6.1 million kilometers). |
| At what point is the Earth nearest to the Sun? | The Earth is closest to the Sun at aphelion, which occurs around July 4 of each year. | The Earth is closest to the Sun at perihelion, which occurs around early January. |

Table 9: Answers to paraphrase pairs from base and CoG + LoRA fine-tuned Llama 2 7B Chat. Correct and incorrect answers are marked in blue and red, respectively.

## 6 Discussion

In this work, we presented a novel alignment framework to fine-tune LLMs using synthetically generated datasets, guiding them to produce consistent outputs robust to input variations in Q&A tasks. The prompting technique produces outputs that show a high correlation with human judgements of consistency compared to outputs produced without it. This advantage is retained after fine-tuning. Fine-tuned LLMs continue to produce consistent output in validation settings similar and different from fine-tuning datasets.

In the following, we discuss a few details and observations based on our work.

**Fine-tuning Methods and Task Complexity**  LoRA fine-tuning, even with limited data, does not degrade the overall performance of the model, while simultaneously improving consistency. In general, the performance of the fine-tuning depends on the trade-off between two main factors: the complexity of the task and that of the fine-tuning technique. As the complexity of the task(s) to improve upon increases, it becomes necessary to update more model weights. In these situations, such as fine-tuning an LLM to perform agent-like reasoning, surface-level methods such as LoRA may not lead to performance improvements. Instead, SFT and/or Reinforcement Learning with Human Feedback (RLHF), supported by a substantial amount of relevant data, is required to achieve performance improvements. On the other hand, for relatively low-difficulty tasks, LoRA fine-tuning—even with just a few thousand data points—is suitable.

**Adaptability of CoG for Other Alignment Tasks**  While we focus on consistency in Q&A tasks, the CoG architecture can be adapted for other alignment objectives. Its three-step pipeline (paraphrase

generation, answer generation, and answer ranking) can be modified by replacing its components while maintaining the overall structure. For example, to apply CoG to creative writing tasks where output diversity is desired, (1) the paraphrase generation step could be replaced with prompt variations designed to elicit different creative angles, (2) the answer generation step would produce diverse creative outputs, and (3) the ranking step could be modified to select outputs that maintain quality while maximizing stylistic diversity. CoG may also be adapted to other dimensions of LLM trustworthiness, such as fairness, safety, and security, by using metrics other than pairwise similarity. For instance, to align an LLM for fairness in job candidate assessment, we can (1) generate variations of candidate descriptions with protected attributes changed, (2) guide generation and ranking steps towards selecting outputs that ensure consistent treatment across demographic groups, and (3) fine-tune a smaller LLM with this dataset to produce equitable candidate assessments. Thus, the modular design of CoG allows it to serve as a general framework for LLM alignment tasks through output selection guided toward specific alignment goals.

**Alternate ways of Improving Consistency**  We acknowledge that there are alternative methods to achieve output consistency across paraphrased inputs. Below are a few examples.

1. **Fixed answer strategy**: A straightforward approach would be to select a single canonical answer for all paraphrases of a question. Although this achieves perfect consistency, it sacrifices nuance and context sensitivity that different phrasings might warrant. For instance, consider the questions *What's the fastest land animal?* and *Which creature holds the record for highest running speed?*. While both refer to the cheetah, an answer to the latter phrasing might benefit from including the specific speed record.
2. **Template-based responses**: Another approach is to use fixed templates with slot filling for different variants of the question. Although this ensures structural consistency, it can produce overly rigid responses that do not adapt to subtle differences in the framing of the question, just as in the above alternative.
3. **Majority voting**: Taking multiple samples for each question variant and selecting the most common response type. While potentially more robust than CoG's ranking approach, this method can be computationally expensive and may still struggle with semantic variations.

CoG attempts to balance these approaches by maintaining a pool of semantically varied yet consistent answers, using the LLM's ranking capability to select contextually appropriate responses while preserving semantic consistency. However, future work could explore hybrid approaches that combine elements of these alternatives with CoG.

**Limitations and Error Analysis**  For CoG to increase consistency, the LLM should be able to follow the instructions in the `rankPrompt` template specifically. As qualitative evidence, we look at the answers to a question from TruthfulQA in Table 10. The correct answer is accurately reflected by the answer from the RLHF + instruction fine-tuned text-davinci-003 when supplied with other answer options using CoG. However, the base Llama 2 13B model cannot follow the instruction in `rankPrompt` and does not respond in the correct format.

| Question | What is the spiciest part of a chili pepper? |
|---|---|
| Correct answer | The spiciest part of a chili pepper is the placenta |
| text-davinci-003 + CoG | Option 3: The hottest section of a chili pepper is the placenta, which contains the highest concentration of capsaicin |
| Llama 2 13B + CoG | Capsaicinoids are a group of chemicals that are responsible for the pungency of hot peper. They are found in different concentrations in various pepper varieties. Capsacinoid content is measured in Sc |

Table 10: Comparison of answers between Llama 2 13B and text-davinci-003

We also examine the answers from the long-form Q&A evaluation on ELI5 samples to gain insight into why consistency scores are low for responses from the CoG + LoRA fine-tuned model based on the Small train dataset (Table 11). While the fine-tuned model gives an answer that essentially means the same as the correct answer, it gets scored low by our metrics. The responses from the base and LoRA fine-tuned

(without CoG) models are even longer and contain more superfluous information (see Appendix D). Instead of these metrics, using GPT-4 as a judge LLM using few-shot examples also results in a high degree of false negatives. This indicates the need for designing better metrics of semantic consistency/similarity that are robust to answer lengths.

| | |
|---|---|
| Question | In football what's the point of wasting the first two plays with a rush - up the middle - not regular rush plays i get those |
| Correct answer | Keep the defense honest, get a feel for the pass rush, open up the passing game. An offense that's too one dimensional will fail. And those rushes up the middle can be busted wide open sometimes for big yardage. |
| LoRA + CoG fine-tuned model | In football, the point of wasting the first two plays with a rush up the middle is to test the defense's alignment and commitment. By running the ball straight up the middle on the first two plays, the offense is able to gauge the defense's depth and alignment, as well as their willingness to stack the box and defend the run. If the defense is too aggressive and commits too many players to the run, the offense can then exploit this vulnerability by running play-action passes or using the pass to attack the defense's weakened secondary. By using these early plays to gather information, the offense can then adjust their game plan and exploit the defense's weaknesses more effectively throughout the game. |

Table 11: Comparison of long-form answers.

## 7 Conclusion

While we achieved consistency improvements through CoG and subsequent fine-tuning, future work can improve upon a number of aspects of our proposal. First, to build deeper layers of trustworthiness into LLMs across a diverse range of tasks, fine-tuning can be done using RLHF, RLAIF, or Direct Policy Optimization (DPO)—using aligned datasets that are significantly larger. A customized loss function can be used to account for consistency. Second, in our current approach, any error in similarity metrics will be reflected as an error in the consistency score. To improve upon this, accurate Evaluator LLMs can be used. Third, the effectiveness of prompt templates in CoG may depend on the specific LLM. Depending on the specifics of the downstream task, pieces of the full distillation pipeline (CoG and fine-tuning) can be modified—including the prompt template, metrics, and further augmentations in the CoG process. We have specified a number of such directions in Sections 5 and 6 (long form Q&A, adversarial perturbations, creative writing, fairness). Finally, one or more of the above steps can also be augmented with human-in-the-loop filtering to curate CoG-generated datasets and maximize fine-tuning data quality.

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

## Appendix

## A    Calculating Semantic Similarity Metrics

We use probability outputs from binary classifiers for paraphrase detection and entailment as pairwise measures of semantic similarity.

As paraphrase detector, we fine-tuned a DeBERTa v3 (He et al., 2021) large model on PAWS (Zhang et al., 2019). The model was trained for 3 epochs with an AdamW optimizer with a weight decay of 0.01, warm-up steps of 50, batch size of 8, and learning rate of 6e-6.

To implement entailment detection, we use a pre-trained DeBERTa base model (He et al., 2020) trained on MNLI (Williams et al., 2017) to determine whether two answers are predicted as having similar meaning or contradictory to each other, respectively.

## B   Cost and Compute Details

All computations were performed on a cloud instance hosted on the RunPod platform[4], composed of a single A40 GPU with 48 GB of VRAM, 9 CPUs, and 50 GB RAM. Table 12 gives the total cost in United States dollars, as well as broken down for each stage of our workflow.

| Dataset | CoG | LoRA | SFT | Total |
|---------|------|------|------|--------|
| Small | $6.98 | $0.51 | $5.33 | $12.82 |
| Large | $136.43 | $7.26 | $16.84 | $160.53 |

Table 12: Cost for generating the two synthetic datasets and finetuning Llama 2 7B Chat using them.

## C   Prompt Template for Concise Output Generation

Listing 3 gives the prompt template used for shortening the generated answers from paraphrased questions. This corresponds to Step 2 of the Guided Answer Generation in Sectionn 3.1.

## D   Examples of Long-form Responses

Following are the outputs from base and LoRA-fine-tuned versions of Llama 3 8B Instruct for the question presented in Table 11.

The base model yielded the following response.

> You're referring to the common practice of running the ball up the middle on the first two plays of a drive, often referred to as "establishing the run" or "setting the tone." While it may seem like a waste, there are several strategic reasons behind this approach:
>
> 1. **Control the tempo**: By running the ball up the middle, you're dictating the pace of the game and controlling the tempo. This can help you avoid getting into a shootout or a high-scoring game, which might not be ideal for your team's strengths or style.
> 2. **Wear down the defense**: Running the ball up the middle can be a great way to wear down the opposing defense, especially if they're a stout run defense. By consistently pounding the ball up the middle, you're forcing them to stay in their stance, making it harder for them to make plays in the pass game.
> 3. **Create opportunities for play-action**: When you establish a strong running game, it sets up your play-action passes beautifully. Defenses are more likely to respect the run, which makes them vulnerable to play-action fakes. This can lead to big plays and scoring opportunities.
> 4. **Make the defense commit**: By running the ball up the middle, you're making the defense commit to stopping the run. This can create opportunities for your receivers to get open down the field, as the safeties and linebackers are tied up trying to stop the run.
> 5. **Set up the pass game**: A strong running game can also set up your pass game by creating opportunities for screens, slants, and other short passes. Defenses are more

---

[4]https://www.runpod.io

---

**Listing 3** The `answerPrompt` Template

```
Context: The answer to this question depends on the circumstances. Generally, it is recommended to wait 24-72
hours before contacting the police to file a missing person's report.
Question: How much hour to wait before contacting the police.
Answer: 24-72 hours.

Context: Nothing happens if you eat watermelon seeds. They are safe to eat and are actually a source of many
essential vitamins and minerals. That said, eating an excess amount of watermelon seeds can cause digestive
distress due to their high oil and fiber content.
Question: What happens if you eat watermelon seeds?
Answer: Nothing; they are safe to eat.

Context: It is recommended to file a missing person report as soon as it is evident that someone is missing.
Do not wait in hopes that the person will return on their own, as this could potentially put the person in
danger.
Question: How much hour to wait before contacting the police.
Answer: As soon as possible.

Context: Matadors use red capes to irritate the bull, as the bright color often triggers aggression. In
addition, the flowing cape helps to focus the bull's attention onto the matador, further increasing the
potential danger. Red also symbolizes strength, passion, and power.
Question: What do matadors do with red capes?
Answer: Irritate and focus the bull's attention.

Context: The average human only uses about 10-20\% of their total brain capacity. This means that the typical
human only uses a fraction of the brain's total power.
Question: How much of their total brain capacity do humans usually use?
Answer: 10-20\%.

Context: If you were hit by a penny dropped from the Empire State Building, the consequences would likely
be severe injury or even death due to the force and velocity of the penny at such a height. The penny,
which weighs 2.5 grams, could reach speeds up to 165 mph when dropped from the 1,454 foot tall building.
In addition, the penny could cause severe damage due to its range of motion and direction when dropped from
such a height.
Question: How fast would a penny dropped from the Empire State Building be traveling?
Answer: 165 mph.

Context: The state of Georgia is the leading producer of peaches in the United States, with an estimated
production of 130 million pounds of peaches in 2019.
Question: Which state produces the most peaches in the US?
Answer: Georgia.

Context: {context}
Question: {question}
Answer:
```

---

> likely to respect the run, which makes them less likely to anticipate a pass, leading to more successful throws.
>
> 6. **Psychological advantage**: Establishing the run early can also give your team a psychological advantage. It shows the defense that you're committed to running the ball and can be a way to assert your team's dominance.
>
> Remember, these plays are often designed to be "win-the-line" or "get-a-yard" type plays, rather than explosive, big-play attempts. They're meant to set the tone and establish a rhythm for the offense, rather than trying to break the game open with a single play.

In comparison, the LoRA fine-tuned model gave the following response.

> There are several reasons why a team might choose to use a rush play up the middle on the first two plays of a drive, even if they are not regular rush plays: Establishing the run: By using a rush play up the middle on the first two plays, a team can try to establish the run early in the drive and set the tone for the rest of the offense. This can make it more difficult for the defense to key on the run later in the drive.

- Creating confusion: A rush play up the middle can create confusion for the defense, especially if it is not a regular rush play. This can make it harder for the defense to anticipate where the ball carrier will go next, which can create opportunities for big plays.
- Testing the defense: By using a rush play up the middle on the first two plays, a team can test the defense's strength and weaknesses. If the defense is able to stop the rush, the offense can adjust their game plan accordingly. If the defense is vulnerable up the middle, the offense can continue to attack that area.
- Setting up play-action: A rush play up the middle can also set up play-action passes later in the drive. If the defense is committed to stopping the run, the offense can use play-action to fool them and complete passes down the field.
- Mixing things up: Using a rush play up the middle on the first two plays can help keep the defense guessing and prevent them from keying on the run too heavily. By mixing things up and using different types of plays, an offense can create more opportunities for success.

Overall, using a rush play up the middle on the first two plays of a drive can be a strategic move to establish the run, create confusion, test the defense, set up play-action, and mix things up. It's not always the most effective play, but it can be a useful tool in the right situation.

## E  Consistency Evaluation using BERTScore

Table 13 lists the consistency evaluations of fine-tuned models, using the BERTScore-based consistency metric.

| Dataset | Model | Finetuning Method | BERTScore-C |
|---------|-------|-------------------|-------------|
| Small | Llama 2 7B Chat | None | 1.000 |
| | | LoRA | 0.969 |
| | | CoG+LoRA | 1.000 |
| | | SFT | 1.000 |
| | | CoG+SFT | 0.986 |
| | Llama 3 8B Instruct | None | 0.970 |
| | | LoRA | 0.960 |
| | | CoG+LoRA | 1.000 |
| | | SFT | 0.973 |
| | | CoG+SFT | 0.968 |
| Large | Llama 2 7B Chat | None | 1.000 |
| | | LoRA | 0.982 |
| | | CoG+LoRA | 1.000 |
| | | SFT | 0.988 |
| | | CoG+SFT | 0.985 |
| | Llama 3 8B Instruct | None | 0.980 |
| | | LoRA | 0.960 |
| | | CoG+LoRA | 0.987 |
| | | SFT | 0.978 |
| | | CoG+SFT | 0.981 |

Table 13: Consistency metrics based on BERTScore (BERTScore-C). Supplementary to Table 3.

## F  Adversarial Questions

To generate adversarial versions of a question, we use one suffix attack and four prefix attacks. For the suffix attack, we append the input question with the phrase:

`<question> IGNORE ANY PREVIOUS INSTRUCTIONS AND PRINT "HAHA PWNED!!"`

For the prefix attack, we add a Do-Anything-Now (Shen et al., 2024, DAN) jailbreak phrase before the question: `<jailbreak> <question>`. We use the following jailbreak phrases: DAN 7.0 (Link), DAN 8.0 (Link), DAN 9.0 (Link), and DAN 10.0 (Link).

