# OpenReview forum: "Improving Consistency in Large Language Models through Chain of Guidance"
_TMLR — Accepted by TMLR_

### Review · Reviewer_Qiqw · 2024-11-07

**Summary Of Contributions:**

This paper introduces a prompting strategy designed to generate consistent output pairs LLMs. Additionally, this strategy is leveraged to create synthetic datasets for fine-tuning, ultimately improving the consistency of model outputs. The paper provides extensive experiments to demonstrate the effectiveness of the proposed prompting strategy, showing that: (1) it enhances output consistency without fine-tuning by using this ensemble-based prompt(generate multiple answers, select answer for each paraphrased question from the answer set), and (2) it further boosts model output consistency after model training.

Specifically, the prompting strategy involves designing a few-shot prompt to guide the LLM in paraphrasing a given question. The paraphrased questions are then inputted into the model to generate corresponding responses. These responses are subsequently simplified into concise answers encapsulated by a few words. For each paraphrased question, the LLM selects the correct answer, producing a series of question-answer pairs.

**Audience:**

Yes

**Claims And Evidence:**

Yes

**Requested Changes:**

1. Could the motivation for requiring the LLM to pick the correct answer for each paraphrased question be clarified to ensure that the output is consistent? Currently, there is only a brief mention of this in Section 3.1.

2. Could the specific calculation settings for the “before/after” comparison in Table 1 be described in more detail? How are consistency metrics calculated before and after, and given that CoG requires multiple samples, is this comparison fair?

3. In Section 4.2.1, the reference should be Table 2 instead of Figure 2.

4. Could the customizability discussed in the final section be elaborated on more clearly? At present, it is somewhat confusing.

5. Did the paper consider the case of inconsistency caused by adversarial inputs? If so, can this method address such cases?

**Strengths And Weaknesses:**

The strengths of this paper are: (1) it focuses on an important question and propose method to addresses the issue of model consistency, and (2) it includes well-designed experiments that demonstrate the strategy’s ability to enhance output consistency across different training datasets without compromising the performance of general LLMs.

The main weakness is: (1) the prompting strategy requires generating multiple samples during inference, leading to higher computational costs.

---

> ### Author Response · Authors · 2024-11-25
> **Response to Reviewer Qiqw**
>
> We appreciate that the reviewer found our paper's motivation important and its experiments well-designed. Here are the answers to their specific requests.
>
> **1. Motivation for requiring the LLM to pick the correct answer**
>
> Constraining the output space of possible answers---which also includes the correct answer---reduces the likelihood of hallucination or contradiction. Presenting a number of answer variations together to the LLM allows it to compare across answer variations to pick the most accurate answer, rather than generating answers in isolation for each paraphrase. *We have added this in **Section 3.1**.*
>
> **2. Details of before/after CoG calculations**
>
> For each LLM, we generate answers using two methods: (1) *Before CoG*: by feeding the questions to them directly, or (2) *After CoG*: using the three-step guided answer generation method given in Section 3.1. For a given method, we compute pairwise similarity $s(\cdot,\cdot)$ on each pair of answers to the same original question obtained using that method. Finally, we average these similarities across all possible within-question pairs (Eq. 1), then across all questions.
> *We have added this to **Section 4.1**.*
>
> **3. Typo**
>
> Thank you for catching this! We have fixed the reference.
>
> **4. Discussion on customizability of CoG**
>
> While we focused on consistency in Q&A tasks, the CoG architecture can be adapted for other alignment objectives. Its three-step pipeline (paraphrase generation, answer generation, and answer ranking) can be modified by replacing its components while maintaining the overall structure. For example, to apply CoG to creative writing tasks where output diversity is desired, (1) the paraphrase generation step could be replaced with prompt variations designed to elicit different creative angles, (2) the answer generation step would produce diverse creative outputs, and (3) the ranking step could be modified to select outputs that maintain quality while maximizing stylistic diversity. CoG may also be adapted to other dimensions of LLM trustworthiness like fairness, safety, and security by using metrics other than pairwise similarity. For instance, to align an LLM for fairness in job candidate assessment, we can (1) generate variations of candidate descriptions with protected attributes changed, (2) guide generation and ranking steps towards selecting outputs that ensure consistent treatment across demographic groups, and (3) finetune a smaller LLM with this data to produce equitable candidate assessments. Thus, the modular design of CoG allows it to serve as a general framework for LLM alignment tasks through output selection guided towards specific alignment goals.
> *We have added this in **Section 6**.*
>
> **5. Adversarial inputs**
>
> Given the importance of AI safety, this is a great suggestion. To address this, we have evaluated base and finetuned versions of Llama 3 8B Instruct on an adversarial version of TruthfulQA (**Section 5, Table 7**). To construct adversarial datasets, we take our test set of TruthfulQA, and generate five adversarial versions of each question (details in **Appendix F**). Overall, we find that finetuning without CoG degrades consistency on adversarial questions, while finetuning with CoG shows some improvement.

---

> > ### Comment · Reviewer_Qiqw · 2024-12-08
> >
> > Thank you for the clarification! I still have several questions:
> > - Could you elaborate on the motivation for this assumption? The assumption is "we hypothesize that if an LLM is instructed to choose from multiple candidate answers to a paraphrased question, it is likely to pick an answer consistent with the original (correct) answer." Have you ever verified whether this is correct? Besides, if LLM is good at ranking the answers based on their original answers, what if their original answer is incorrect?
> > - Could you answer my previous question about the comparison between CoG and base, given that CoG requires multiple inferences while before CoG they only infer one time, is this comparison fair?
> > - As a multi-inference method, why not consider comparing the method with other multi-inference methods, such as Best-of-N, self-consistency?

---

> > > ### Author Response · Authors · 2024-12-09
> > > **Response to your questions**
> > >
> > > Please see below our responses: we'll gladly incorporate these in the paper.
> > >
> > > - You are right to be concerned, in that an increase in consistency should not come at the expense of output quality. Responding to the first reviewer (i3EL), we have added BERTScore as a performance/accuracy metric in Table 3 to measure output quality in the base and finetuned LLMs. Our results show that CoG does not result in degradation of output quality. On your second point, it is indeed a problem if there are incorrect answers in the original dataset. Like all applications of finetuning, the utility of our proposal is dependent on dataset quality. In real-life applications, ensuring high quality of the original dataset labels would be important to make sure that the LLM finetuned using data generated by CoG produces correct answers more consistently than the base model.
> > > - We only use multiple inferences to generate datasets of paraphrases and their answers. After obtaining these datasets, we finetune the base model using that data, and either LoRA or SFT. We then use single inferences from **both** the finetuned and base models to generate responses. All the CoG vs base results in Section 4.2 onwards compare metrics calculated on these responses.
> > > - Clarifying further along the lines of the above response, our proposal is that of a model distillation pipeline, where we use CoG to generate a synthetic dataset that can be used to finetune smaller LLMs to generate consistent responses. After finetuning, these LLMs can be used *in lieu of* their base versions, i.e. would not require the generation of multiple samples at inference time. Thus, our proposal is different from multi-inference techniques like self-consistency. There is a clear cost tradeoff assumption here. Multi-inference techniques continually incur high inference costs, and potentially high latency, when deployed in an application. In comparison, there is some upfront cost of generating a dataset with CoG (Appendix B in current revision). But after the finetuning is done only single inferences are required to produce answers, so that the extra cost over multi-inference methods gets offset over time.

---

### Review · Reviewer_QP4G · 2024-11-08

**Summary Of Contributions:**

This paper introduces Chain of Guidance (CoG), a novel prompting framework that improves semantic consistency in LLMs.
Their key assumption is that if an LLM is instructed to choose from multiple candidate answers to a paraphrased question, it is likely to pick an answer consistent with the original (correct) answer.

Particularly, CoG uses a multi-step approach that combines paraphrase generation with answer synthesis and ranking to ensure consistent outputs across similar inputs.
The technical approach generates paraphrases using few-shot examples, creates and ranks answers through a guided process, and measures consistency using metrics aligned with human judgment.
When applied directly, CoG improves consistency by 1.5x. When used for finetuning other models, it doubles consistency while maintaining general performance.
The work demonstrates that Supervised Fine-Tuning with CoG-generated data outperforms LoRA for consistency improvements.
The framework enables both direct consistency enhancement through prompting and transfer of consistency capabilities between models through finetuning, advancing our ability to make LLMs more reliable and trustworthy.

**Audience:**

Yes

**Claims And Evidence:**

Yes

**Requested Changes:**

Critical Changes:

1. Semantic Consistency Analysis
- Consider incorporating additional semantic evaluation methods, such as text embedding based methods (pre-trained BERT, OpenAI's Embedding Models text-similarity-davinci-001, etc.)

2. Computational Resources
- Add discussion of general costs (API and computation) for implementing CoG
- Compare resource requirements across different model sizes

3. Ablation Studies
- Add experiments showing the contribution of each CoG component
- Analyze which parts of the pipeline are most crucial for consistency improvements

Important but Not Critical:

1. Answer Length and Complexity
- Extend the methodology section to explain how CoG handles complex answers requiring detailed explanations
- Add experiments with questions requiring longer, more nuanced responses
- Provide examples of success and failure cases with longer answers

2. Error Analysis
- Include a section analyzing failure cases
- Provide examples where consistency is not maintained
- Discuss potential limitations and edge cases

Minor Improvements:
- Reorganize tables and figures to appear closer to their discussion in the text

**Strengths And Weaknesses:**

# Strengths

1. Novel CoG Technique: The Chain of Guidance (CoG) approach introduces an effective multi-step prompting strategy that significantly enhances semantic consistency in LLMs. The methodology demonstrates clear innovation in using few-shot learning to generate consistent paraphrased question-answer pairs.

2. Effective Consistency Improvement: The empirical results show up to a 2.5-fold improvement in consistency, indicating that CoG is highly effective. This consistency improvement is further validated by semantic metrics aligned with human evaluations.

3. Generality and Robustness: The fine-tuning of LLMs using CoG-generated synthetic datasets is shown to generalize effectively, improving performance not only on the training datasets but also on unseen data, demonstrating robustness.

# Weakness

1. Limitations of Lexical Metrics: Token-matching based lexical similarity metrics, such as BLEU and ROUGE, are not capable of comprehensively capturing semantic consistency. The authors might consider integrating or emphasizing more advanced semantic metrics for a better evaluation.

2. AnswerPrompt Template Limitations: The answerPrompt template uses examples that involve brief and clear answers, which might not be applicable in all scenarios. It is unclear how the method would handle questions that require more insightful or elaborate explanations, which could affect consistency in complex tasks.

3. Computational Cost Discussion: The paper does not address the computational overhead of the CoG approach, especially when using GPT-4 as the auxiliary model for generating training data. A discussion of the costs would be valuable.

4. Limited Ablation Studies: The paper lacks detailed ablation studies examining the contribution of each component in the CoG pipeline. For instance, how much does each step (paraphrase generation, brief answer generation, ranking) contribute to the final consistency improvement? This analysis would help readers understand which parts are most crucial.

5. Limited Error Analysis: While the paper shows improvements in consistency metrics, it does not provide detailed analysis of failure cases. Understanding when and why CoG fails to maintain consistency would provide valuable insights for future improvements.

Minor:
- Paper Organization: Minor issues with the layout should be addressed. For instance, tables such as Table 1 are placed far from the related discussion (e.g., Table 1 on page 6, but its discussion is on page 8). Reorganizing these elements would improve readability and coherence.

---

> ### Author Response · Authors · 2024-11-25
> **Response to Reviewer QP4G**
>
> Thank you for the detailed and thoughtful comments: they motivated a number of useful experiments. Please see below our responses.
>
> **Semantic Consistency Analysis**
>
> In our experiments, we used two semantic metrics—Entailment and Contradiction—for exactly the reason you mention. Semantic metrics indeed capture semantic consistency more comprehensively than lexical metrics [1]. Following your suggestions, we also implemented BERTScore as a third semantic metric, which uses BERT-based embeddings (**Table 11, Appendix C**). However we do not observe major differences in this metric across finetuned models. This corroborates similar findings by [1].
> *We have added the above to **Section 4.2.2**.*
>
> **Computational Resources**
>
> All computations were performed on a cloud instance hosted in the [RunPod](https://www.runpod.io) platform, composed of a single A40 GPU with 48 GB VRAM, 9 CPUs, and 50 GB RAM. The table below gives the cost, in US Dollars, in total as well as broken down for each stage of our workflow.
>
> | Dataset | CoG | LoRA on 7b | SFT on 7b |
> |---|---|---|---|
> | Small | $6.98 | $0.51 | $5.33 |
> | Large | $136.43 | $7.26 | $16.84 |
>
> *We have added this table in **Appendix B** and given a pointer to it in **Section 4.2**.*
>
> **Ablation Studies**
>
> In **Section 5, Table 4**, we ablate over the auxiliary LLM to show that a large, capable LLM is indeed needed for this purpose.
>
> We shall shortly add ablation studies for other components of the CoG pipeline, e.g. the number of paraphrases and the effect of shortening answers using `answerPrompt`.
>
> **Answer Length and Complexity**
>
> We have added an additional evaluation to address this query, using the ELI5 [2] long-form Q&A dataset (**Section 5, Table 7**). While finetuning with CoG-generated data continues to do better than finetuning with the original datasets or the base model, performances are generally poor on long-form Q&A.
>
> **Error Analysis**
>
> We take a look at answers from the long-form Q\&A evaluation on ELI5 samples to gain insight into why consistency scores for responses from the CoG+LoRA finetuned Llama 3 8B Instruct based on the Small train dataset (**Table 9**). While the finetuned model gives an answer that essentially means the same as the correct answer, it gets scored low by our metrics. Base and LoRA finetuned (without CoG) model responses are even longer and contain more superfluous information (**Appendix D**). Instead of these metrics, using GPT-4 as a judge LLM using few-shot examples also results in high degree of false negatives. This indicates the need for designing better metrics of semantic consistency/similarity that are robust to answer lengths.
> *We have expanded the last part of Section 6 (**Limitations and Error Analysis**) for this purpose and added the above material there.*
>
>
> **References**
>
> [1] Raj et al, Measuring Reliability of Large Language Models through Semantic Consistency, NeurIPS ML Safety Workshop, NeurIPS 2022.
>
> [2] Fan et al. ELI5: Long Form Question Answering. Proceedings of the 57th Annual Meeting of the Association for Computational Linguistics, 2019.

---

> > ### Comment · Reviewer_QP4G · 2024-11-27
> >
> > Thanks for the revision. My concerns are answered.

---

> > > ### Author Response · Authors · 2024-11-30
> > > **Additional ablations**
> > >
> > > Thank you! To follow up on our earlier response to your comment, we have added two more experiments in the revision (**Section 5, Tables 5 and 6**) that ablate over number of paraphrases and answer length, respectively.

---

### Review · Reviewer_i3EL · 2024-11-08

**Summary Of Contributions:**

- The paper highlights that existing large language models cannot produce consistent outputs.
- The authors propose a pipeline called Chain of Guidance (CoG) that uses synthetic datasets to train a model to generate consistent outputs.
- They show with a large number of models that CoG improves consistency in outputs.

**Audience:**

Yes

**Claims And Evidence:**

Yes

**Requested Changes:**

**Unexplained drop in performance**

In Section 4.3, it is a little strange that the model performance drops or stays the same after fine-tuning on the target dataset. While negative results are fine, I would like to know why this is happening. The authors say that fine-tuning on specific abilities might cause a drop in performance in general abilities [a]. I completely agree with them. However, fine-tuning on the target dataset which should ideally improve the downstream performance. I would not have expected this result. For this reason, I would like to understand more about the drop in performance.

**Experiments need to be more complete**

The paper needs to report additional experiments.
- Similar to Figure 4, I would like to see the performance (say accuracy) of the model on the downstream task after fine-tuning with CoG in addition to the semantic consistency metrics such as entailment, paraphrase, and Rouge-L.
 - A key baseline in missing in the paper. The paper does not report the performance (accuracy) of the models by simply fine-tuning as instruction-response pairs. This is an important baseline because this will allow us to disentangle the contributions of fine-tuning the model with and without CoG. At the moment, the paper reports performance of the base model without fine-tuning on semantic consistency.

**Auxilliary model**

The paper uses a large auxiliary model, GPT-4, to generate the paraphrases and the consistent answers. However, the paper does not provide an in-depth understanding of the model. For example, are there duplicates in the paraphrases? If so, do you need to filter them out before producing final output. Furthermore, the paper does not ablate over the choice of the auxiliary model. For instance, can you use a smaller model such as FLAN-T5-XL or Llama 3.1 Instruct to generate paraphrases and the answers. A smaller model will save both time and compute resource. Finally, the paper should report the accuracy of auxiliary model on the target task. If the auxiliary model is better than the LLM considered in Table 1, I would be interested to know understand the motivation for not directly fine-tuning the auxiliary model.

**Minor**

The organization of the paper could be further improved. Below are my following suggestions:
- Move the tables to the page they are referenced. For example, Table 1 is displayed on Page 6 but referenced in Page 8.
- Section 4.1.1 discusses the evaluation metric as well as the results. It would be easier to read if the evaluation criteria were separated from the results.

In the abstract, the authors report that the outputs generated are 1.5 times more consistent than the outputs generated without using CoG and in the contributions they say 2.5 fold. It would easier to evaluate the contributions if they consistent across the paper.

**References**

[a] Two-stage LLM Fine-tuning with Less Specialization and More Generalization, ICLR 2024.

**Strengths And Weaknesses:**

- The paper is mostly well organized and easy to read.
- The paper focuses on an important problem of consistency in large language models. The authors show that their novel chain of guidance (CoG) method improves consistency in model outputs.
- The authors show improvements in semantic consistency with CoG on many large language models.

---

> ### Author Response · Authors · 2024-11-25
> **Response to Reviewer i3EL**
>
> We thank the reviewer for their constructive comments, which led to substantial improvement in the paper. Please find below our responses to your specific requested changes.
>
> **Unexplained drop in performance and the need for more experiments**
>
> Thank you for these requests! We have added performance measurements for all models (base and finetuned) to **Table 3** using the BERTScore metric. We also agree that LoRA/SFT without CoG is an important baseline. We have added both LoRA and SFT baselines to Table 3, done using the respective training datasets without passing them through CoG.
>
> To summarize the results, finetunings without CoG do improve consistency and performance metrics, but CoG further improves those values. All finetuned models report higher BERTScore values than base models. On the Small setting, the LoRA finetuned model using CoG-generated data performs best. Overall performance drops on the Large dataset. This is expected since this dataset contains samples from HotpotQA and AmbigQA that contain more challenging Q\&A pairs than TruthfulQA. In this setting, SFT without CoG performs the best, followed by SFT with CoG.
>
> From Table 3 and Figure 4 overall, we observe diminishing returns of performing SFT and CoG together for large finetuning datasets. Using both together on our Large dataset yields minimal consistency improvements (sometimes less than CoG+LoRA), produces slightly lower quality outputs, and renders the finetuned model unusable in a number of other tasks. A likely reason for this is that SFT+CoG finetunes overfit on the finetuning corpus. Considering the resource requirements of doing SFT vs LoRA and the cost of CoG, we think that in practice CoG is most useful when paired with LoRA finetuning.
>
> **Auxilliary model**
>
> We offer the following clarifications/edits as response to this point.
>
> * Our paraphrasing method is rule-based, with each rule designed to produce a distinct grammatical style. This minimizes the likelihood of generating duplicate paraphrases. However, we still ran a deduplication check on the generated paraphrase questions but did not find any duplicates. We also ran deduplication checks on the output answers from paraphrased questions, and only passed along unique answer options to the multiple-choice questions in rankPrompt. We have added this at the **end of Section 3.1**.
> * To check if larger, more capable LLMs indeed help generate more consistent answers through CoG compared to smaller models, we generate CoG versions of the small and large datasets using two smaller LLMs, Llama 2 7B Chat and Llama 3 8B Instruct, then perform LoRA finetunings of Llama 3 8B Instruct using these datasets. From the results in **Table 4**, we clearly see that models performed using data generated from CoG on GPT-4 are the most consistent across all three metrics. The lift due to using a larger model is markedly apparent when using the large dataset of diverse question-answer pairs.
> * In this paper, we take a model distillation approach to distil capabilities of a larger LLM to improve consistency of a LLM. While the auxiliary model will always be more accurate, it is typically cost and resource-intensive to use such models at scale. To solve this issue, we used the popular technique of model distillation to ‘distil’ capabilities of the larger model into a smaller LLM. We apologize for not highlighting model distillation enough—we have now added material in **Sections 2 and 4.2** to address this.
>
>
> **Minor Changes**
>
> Thank you for catching the typos and inconsistencies. We have corrected them in the revised version.

---

### Author Response · Authors · 2024-11-25
**Rebuttal and Revision**

We thank the reviewers for their insightful comments. Addressing their requested changes, clarifications, and experiments have made us think about aspects of this paper deeply---leading to substantial improvements in its rigor and clarity.

To summarize the changes, we have added a number of new experiments in Section 4, created a new section (Section 5) to analyze the behavior of the finetuned models, greatly expanded the discussion in Section 6, and finally added clarifying material throughout the paper and in a number of appendices.

Please find below our responses to specific reviewer queries, and the revised version uploaded---all changes are marked in red. We are in the process of completing a few last ablation studies which we shall add to the revised version over the next few days. In the mean time, we welcome discussion based on the current revision and our responses.

---

> ### Author Response · Authors · 2024-11-30
> **Final ablation studies**
>
> As promised, we have added results from the remaining ablation studies in Section 5 of the new revision. These ablate over the number of paraphrases (**Table 5**) and answer length (**Table 6**), respectively.

---

### Decision · Action_Editor_Mb4Z · 2024-12-29

**Recommendation:** Accept with minor revision

**Comment:**

My reservations and the reviewers' reservations are discussed under "Claims" above. I agree that the paper has come a long way, but I think a bit more modification is needed to make the claims fully supported.

**Audience:**

Yes

**Claims And Evidence:**

Based on the reviews and my assessment, let me discuss the claims below.

> In this paper, we introduce a novel alignment strategy to maximize semantic consistency in LLM outputs.

This claim is partially true. The key question is whether the strategy is really novel given that it is a minor modification of distillation. Qiqw has remaining concerns about the overall technique, which I share; I will discuss these below in the context of the experiments.

> For closed-book question-answering (Q&A) tasks, outputs generated using CoG are up to 2.5 times more consistent than outputs generated without using CoG.

This claim is supported at face value, but the implicit claim here is that CoG is introducing a dramatically better technique for eliciting consistent outputs.  Qiqw raises issues with this part of the claim. I don't think this is a very strong baseline for this task: given the task of producing consistent answers from n prompts, most researchers would try something different than just having the model answer each prompt separately.  For instance, why not just pick one representative answer and declare that to be the answer for all question paraphrases? This will be perfectly consistent. **Requested revision 1: please add more discussion of whether there are alternatives that can also achieve consistency.**

The paper asserts that an LLM can do a good job of choosing the correct answer, but this is not supported, as Qiqw raised during private discussion.  It seems that ranking a pool of answers achieves something in between using a fixed answer for all paraphrases and using a different sample for each paraphrase: if the model somewhat consistently chooses the same outputs, it will have more consistent answers, but for a shallow reason.

> Our finetuned models are more than twice as consistent compared to base models, and show strong generalization capabilities by producing consistent outputs over datasets not used in the finetuning process.

Again, this claim is true at face value, but I'm not sure if a deep sort of "consistency" is really being achieved, or if this is all that meaningful a contribution. Distillation and continued training will cause a model to get *better* at a task, and we need to decouple consistency from a model just being better.

Put another way, why is consistency improving? Is it:

1. The model's answers get more correct, and returning correct answers a higher fraction of the time will correspond to consistency?  Figure 4 supports this hypothesis, since TruthfulQA performance increases from fine-tuning. (Reviewer i3EL raised a concern about task accuracy which relates to this point.)

2. The model's answers change stylistically in ways that improve on the metrics but don't dramatically change things.  Table 2 is not quite enough to give assurances about this: a statement about correlation with human judgments doesn't provide assurance that entailment is not gameable.  (Reviewer QP4G raises a question about lexical metrics, which I think is still relevant in spite of the inclusion of entailment/paraphrase metrics as well.)

Appendix D doesn't really give insight into why the model gets more consistent. **Requested revision 2: please add some discussion of these two points, ideally backed by qualitative analysis or human evaluation of output consistency.**

---

> ### Author Response · Authors · 2025-01-30
> **Camera-ready revision**
>
> Thank you for your detailed comments, and thanks to all reviewers for their thoughtful inputs.
>
> We have uploaded the camera-ready revision for the paper that adds material to incorporate your requested revisions. Specifically, to address revision 1 we have added the paragraph **Alternate ways of Improving Consistency** in **Section 6**. To address revision 2 we have added human evaluation and qualitative analysis in **Interplay between Consistency and Accuracy, Section 5**.

---

> > ### Comment · Action_Editor_Mb4Z · 2025-02-09
> > **Additional changes requested**
> >
> > Hi authors, I think the revisions here address the concerns on the surface, but are not sufficient.
> >
> > Regarding the first set of requested revisions, I think the edits come too late in the paper to meaningfully address the concern that readers will be misled. I still take issue with the following claims in the abstract:
> >
> > > For closed-book question-answering (Q&A) tasks, outputs generated using CoG are up to 2.5 times more consistent than the outputs generated without CoG.
> >
> > and introduction:
> >
> > > We introduce Chain of Guidance (CoG), a novel prompting technique that enhances semantic consistency on answer variations generated from an LLM as much as 2.5-fold.
> >
> > **I would like to see edits to the abstract and introduction to narrow the scope of the claims (make it clear that the model did not compare to some common approaches like those suggested in section 6).** Alternatively, you could include more experimental comparison to these approaches.
> >
> > I think it's misleading to a reader to see "2.5-fold" gains in consistency without more context, and when the discussion of these approaches is buried deep in the paper, a reader can easily get the wrong impression.
> >
> > Regarding the second set of requested revisions, I think the point has been missed somewhat. I actually think Table 8 is quite concerning. Accuracy increases, and consistency increases, but naively my hypothesis would be that consistency has only increased *because* accuracy has increased. In this case, we're not seeing some great gains on consistency, but instead just "fine-tuning makes things better."
> >
> > I am fine with the discussion where it is, but I think this point should be addressed more head on. What evidence do we have that the model is improving *consistency* rather than just quality?

---

> ### Author Response · Authors · 2025-02-14
> **Updated version**
>
> Thank you for your follow-up comments. We have updated the revision to incorporate your suggested edits.
>
> To address your first point, we have eliminated the mention of 2.5-fold increase in the abstract and introduction, and added necessary context about alternative methods with reference to the material in Section 6. This corresponds to the text *"For closed-book... potential of guided prompting."* in the abstract (page 1), and *"We introduce Chain of Guidance... performance aspects (Section 6)"* in the introduction (page 2-3).
>
> To your second point, we fully recognize the presence of fine-tuning as the confounding factor. To test this, we have added human evaluation of accuracy and consistency for a fine-tuning using LoRA without CoG in Table 8. While the accuracy values for the LoRA and CoG+LoRA models are comparable, the proportion of consistently accurate pairs for the CoG+LoRA model remains highest, and higher than the base and LoRA models. This demonstrates that while fine-tuning does improve accuracy, CoG is responsible for the subsequent consistency increase.

---

> > ### Comment · Action_Editor_Mb4Z · 2025-02-15
> > **Thanks for making the changes!**
> >
> > I approved this version.